# Dharma: A novel, clinically grounded machine learning framework for pediatric appendicitis—Diagnosis, severity assessment and evidence-based clinical decision support

Anup Thapa Kshetri[1]*, Subash Pahari[2], Shashank Timilsina[1], Binay Chapagain[3]

**1** Department of Emergency Medicine and General Practice, Matri-Sishu Miteri Provincial Hospital, Pokhara, Gandaki, Nepal, **2** Department of Computer Engineering, Faculty of Science and Technology, Pokhara University, Pokhara, Gandaki, Nepal, **3** Department of Critical Care Medicine, Tribhuvan University Teaching Hospital, Kathmandu, Bagmati, Nepal

* ajungthapa11@gmail.com

## Abstract

Acute appendicitis is a common but diagnostically challenging surgical emergency in children. Existing linear scoring systems lack sufficient accuracy for standalone use, while advanced imaging is constrained by risks of sedation, contrast, and radiation. Furthermore, no available tools provide prognostic guidance. We introduce *Dharma*, a machine learning framework consisting of a clinically grounded imputer and two random forest classifiers for diagnosis and severity assessment. Designed for real-world bedside use, Dharma is open-sourced and accessible through a web application. Dharma achieved excellent diagnostic performance, with an AUC-ROC of 0.98 [0.97–0.99] and accuracy of 93% [91–95]. For prognostic classification, it identified complicated appendicitis with high sensitivity (96% [93–99]) and negative predictive value (97% [94–99]). Even in cases without appendix visualization—a frequent limitation in resource-constrained settings—Dharma maintained strong performance (AUC-ROC 0.96 [0.93–0.99]), with specificity of 97% [93–100] and PPV of 93% [84–100] at a 44% threshold, and sensitivity of 92% [84–98] with NPV of 95% [91–99] at a 25% threshold. These threshold-dependent trade-offs enable Dharma to support both ruling in and ruling out appendicitis within diverse clinical workflows. Beyond pediatric appendicitis, Dharma's open-source framework and clinically grounded design also provide a generalizable foundation for developing equitable and practical decision-support systems in healthcare.

## Author summary

Accurate diagnosis and risk stratification of pediatric appendicitis remain challenging due to heterogeneous clinical presentations, the absence of definitive

**Data availability statement:** We used the publicly available dataset described in the Data Source section of the manuscript. All original datasets, as well as the subsets and combinations we generated, are accessible in the public repository: https://github.com/ajung17/Dharma-AppendicitisModel.

**Funding:** The author(s) received no specific funding for this work.

**Competing interests:** The authors have declared that no competing interests exist.

biomarkers, and limited access to reliable imaging, particularly in resource-constrained settings. Existing clinical scoring systems, while simple and widely used, rely on fixed linear assumptions, lack interpretability at the individual patient level, and are restricted to diagnostic decision-making without addressing disease severity or prognosis.

To address these limitations, we developed **Dharma**, a clinically grounded, interpretable machine learning-based framework designed to support both diagnosis and severity assessment of pediatric appendicitis using routinely available clinical, laboratory, and radiological data. Rather than replacing clinical judgment, Dharma is intentionally designed to mirror real-world clinical reasoning, explicitly model non-linear interactions among features, and provide transparent explanations for its predictions.

Implemented as an open-access, web-based clinical decision support tool, Dharma delivers real-time, evidence-based risk estimates that are adaptable to varying resource settings. Beyond pediatric appendicitis, this work demonstrates a generalizable, medicine-first approach to developing equitable and interpretable AI systems that complement clinician decision-making and align with the evolving needs of 21st-century healthcare.

## Background

Acute abdominal pain (AAP) is a common complaint of Pediatric Patients which accounts for up to 10% of total visits to the Emergency Department while acute appendicitis is the major cause of AAP in children over the age of 1 [1,2]. Despite being a routinely encountered clinical dilemma, acute appendicitis remains a challenging diagnosis for healthcare professionals. This is likely due to variability in clinical presentation and the frequent absence of classic symptoms, especially in children. Various scoring systems, with Alvarado being the oldest and most widely used, help physicians quantify the risk of pediatric appendicitis. However, none of the scoring systems provide sufficient sensitivity, specificity or predictive values to be used as an exclusive standard in setting the diagnosis. [3–5].

The increasing reliance on CT and USG for evaluating acute abdominal pain has not been accompanied by a corresponding improvement in appendicitis detection rates [6]. Misdiagnosis rates remain alarmingly high, ranging from 70% to 100% in children aged three years or younger, with a gradual decrease as age increases [7]. This diagnostic challenge is often attributed to the atypical presentation of the disease in younger children and the anatomical variability of the diameter of the appendix. Consequently, the lack of a reliable diagnostic tool may contribute to the persistently high incidence of Negative Appendectomy (NA) worldwide, reported between 5% and 20% in various centers [8–11]. NA is a dreaded outcome for any operating surgeon due to the inherent risks associated with general anesthesia and surgery, particularly in the pediatric population.

Moreover, no traditional inflammatory biomarkers, clinical signs and symptoms, or imaging modalities can reliably predict the progression of uncomplicated to complicated appendicitis [12,13]. This is particularly concerning in pediatric patients, where the incidence of perforated appendix is alarmingly high and increases as the age decreases, reaching over 80% in children younger than three years old [14]. This highlights the urgent need to refine diagnostic and prognostic strategies for pediatric appendicitis.

Machine Learning (ML), a subfield of Artificial Intelligence (AI), involves learning patterns from data and applying this knowledge to make predictions in future scenarios [15]. In medicine, ML offers the potential to enhance diagnostic, prognostic, and management precision by integrating heterogeneous clinical, laboratory, and imaging data into predictive frameworks. Accordingly, AI and ML have seen growing application across medical domains, with recent emphasis on Large Language Models (LLMs) and other data-driven architectures.

However, in high-stakes clinical settings, predictive performance alone is insufficient. Clinical decision-making occurs under uncertainty, incomplete information, and asymmetric risk, where the consequences of false negatives and false positives differ substantially. As a result, AI systems are most effective when they are designed around clinical reasoning, workflow constraints, and patient safety considerations, with ML serving as a supporting component rather than the primary driver. Hybrid, human-centered approaches—where model behavior, interpretability, and optimization objectives are explicitly aligned with clinical priorities—are more likely to generalize, earn clinician trust, and operate safely in real-world environments [16–18].

Placing human understanding of disease progression, decision pathways, and clinical constraints at the core of system design, while leveraging ML as an enabling tool, may better realize AI's potential to deliver accurate, efficient, and accessible healthcare globally [18,19].

ML has been applied to appendicitis diagnosis since the 1990s, with pediatric-specific models emerging in the early 2010s [20]. These models have primarily focused on predicting pediatric appendicitis, associated complications, and post-operative recovery outcomes [21,20,22–28]. Despite the development of numerous ML-based tools reporting promising retrospective performance, none have been widely adopted as real-world clinical decision support systems.

ML-based approaches have the potential to enhance both diagnostic accuracy and severity prediction in pediatric appendicitis [29], thereby reducing unnecessary radiation exposure from computed tomography, minimizing sedation-related risks, lowering negative appendectomy rates, and optimizing referral pathways from primary care. However, achieving these benefits requires models that go beyond retrospective performance metrics and demonstrate real-time clinical utility [18]. Such systems must perform reliably even with missing or incomplete data and remain applicable across diverse healthcare environments, from well-resourced urban hospitals to resource-limited rural settings. Therefore, there is a pressing need for a transparent, interpretable, adaptable, and user-friendly clinical decision support system that can reliably assist in diagnosis, rule out disease, stratify complication risk, and guide management decisions in pediatric appendicitis.

Thus, we defined two primary objectives for this study:

- To develop and evaluate a clinically informed, optimized, and generalizable ML framework for diagnosing pediatric appendicitis and predicting complications, and to compare its performance with conventional diagnostic tools (AS, PAS, AIR, Tzanaki, USG) as well as state-of-the-art ML architectures (XGBoost, LightGBM).

- Deploy the ML framework as an interpretable, web-based diagnostic and prognostic tool that facilitates evidence-based clinical decision-making for pediatric appendicitis, with particular applicability in resource-constrained settings.

## Materials and methods

### Ethics statement

The original dataset was collected in accordance with the ethical guidelines and approval procedures of the University of Regensburg(approval numbers: 18-1063-101, 18–1063_1–101, and 18–1063_2–101). For the present study, only

anonymized secondary data were analyzed. In line with these institutional regulations, no identifiable patient information was included, and therefore additional ethical approval and individual informed consent were not required.

## Data source

This study analyzed anonymized secondary data from pediatric patients with suspected appendicitis. In the primary cohort, diagnosis data were available for 780 of 782 patients admitted with abdominal pain to Children's Hospital St. Hedwig in Regensburg, Germany, between 2016 and 2021 [30]. For testing the diagnostic model, 289 unique records from 430 suspected appendicitis cases collected at the same hospital between January 1, 2016, and December 31, 2018, were used [21]. To develop the severity assessment model, the training dataset was augmented with 49 complicated cases from 301 records collected between 2015 and 2022 at the Department of Pediatric Surgery and Pediatric Traumatology, Florence-Nightingale Hospital, Düsseldorf, Germany [31]. All datasets were already anonymized, with patient identifiers removed, and are publicly accessible from the respective sources.

## Dataset description

The primary dataset consisted of 782 suspected pediatric appendicitis cases collected over a three-year period at a tertiary children's hospital in Germany [30]. Each record contained 55 demographic, clinical, laboratory, and radiological variables, along with labels for diagnosis, treatment modality, and severity. The cohort included children and adolescents aged 0–18 years who presented with abdominal pain and were evaluated for suspected appendicitis. Patients with a prior appendectomy, concurrent abdominal pathologies, or pre-evaluation antibiotic use for unrelated infections were not included. After applying these criteria and excluding two records lacking diagnostic information, 780 cases with confirmed diagnostic labels were available for analysis.

- Diagnosis: appendicitis (n = 463) and no appendicitis (n = 317)

- Severity: complications (n = 119) and no complications or no appendicitis (n = 661)

- Treatment Modality: conservative (n = 483) and surgical (n = 297)

For severity assessment modeling, the primary dataset was augmented with 289 additional unique records from the cohort used in literature [21], yielding a combined dataset of 1,069 patients after deduplication. Only patients with confirmed appendicitis were retained for prognostic modeling, resulting in a final sample of 650 cases, of which 490 had uncomplicated disease and 160 presented with complications. This subset of the dataset allowed us to train and evaluate a clinically meaningful model focused specifically on prognostic discrimination following diagnostic classification.

Given the scarcity of publicly available pediatric appendicitis datasets, we used all curated cases from the three high-quality cohorts [21,30,31], applying task-specific inclusion criteria to derive the diagnostic and prognostic subsets used in this study, as described in detail in the Model Development section.

## Data preprocessing

Data preprocessing involved standardizing categorical encodings: binary variables were mapped to 0/1, urinary ketones were coded as 0 (absence/trace), 1 (1+), 2 (2+), and 3 (3+), peritonitis was coded as 0 (absent), 1 (local), and 2 (generalized), and stool changes were coded as 0 (normal) and 1 (any alteration, including diarrhea or constipation).

Missing values were present in almost all variables (see S1 Table). The appendix diameter variable had 36% missingness, yet it was retained because of both the inherent challenges of appendix visualization on ultrasonography (USG) and its well-established diagnostic and pathophysiological relevance in appendicitis.

For initial exploratory analyses, statistical tests were conducted on the available data without imputation. For model development, missing values in selected predictors were handled using a custom imputation pipeline, the Dharma_Imputer. This approach incorporated domain knowledge by grouping inflammatory markers (WBC count, neutrophil percentage, CRP, and

temperature) and imputing them through iterative imputation with scikit-learn's decision tree regressor. Categorical variables were then imputed separately, with each feature modeled by a dedicated decision tree classifier. The appendix diameter feature was treated distinctly: missing values were flagged by an indicator column and imputed with a clinically impossible placeholder value of -1, reflecting that non-visualization of the appendix on ultrasound is common and diagnostically meaningful. This strategy preserved the clinical nuance that decision-making differs depending on whether the appendix is visualized or not.

To preserve data integrity, imputations for training, validation, and test sets were performed independently. The trained Dharma_Imputer was also integrated into the web-based clinical decision support tool, enabling clinically informed, real-time handling of missing values.

For the primary classification task, we used the full dataset of 780 cases. Additionally, 239 unique cases from the 430 suspected appendicitis cases [21] were used exclusively for external evaluation of our diagnostic models and existing linear scoring systems.

For the severity assessment task, the dataset was augmented with the full set of 430 suspected cases [21]. After duplicate removal, the combined dataset comprised 1,069 unique patient records. Among these, only patients with a confirmed diagnosis of appendicitis (n = 650) were included for training and testing the severity assessment model.

Treatment modality was not analyzed in this study, as we consider it a clinical decision to be determined by the operating surgeon based on the diagnosis and anticipated risk of complications.

## Feature selection

Advanced radiological features such as fecal impaction, bowel wall thickening, or pathological lymph nodes were excluded from the study, as these findings can be challenging to assess even for experienced radiologists. Our aim was to develop a model that could augment pediatric surgical decision-making, particularly in resource-constrained settings where access to advanced radiology is almost always limited.

The Chi-Square test was performed to assess the association between categorical variables and disease state, while Cramér's V measured the strength of these associations. Variables with a significant association ($p < 0.001$) and a moderate to strong effect size (Cramér's $V > 0.10$) were selected for the development of the model. Contralateral rebound tenderness, despite meeting the initial inclusion criteria, was excluded from the model's feature pool as it is not a classically defined or standardized clinical sign in the diagnostic framework of acute appendicitis.

For continuous variables, the Mann–Whitney U test was applied to compare distributions between positive and negative classes, and rank-biserial correlation was calculated to quantify effect size. Features with significant differences ($p < 0.001$) and correlation coefficients $> 0.40$ were retained for the feature pool.

All statistical analyses were conducted in Python using Pandas, NumPy, and SciPy on the original dataset of 780 suspected appendicitis cases.

To refine feature selection, Recursive Feature Elimination with Cross-Validation (RFECV) was employed to identify the most influential features. Different subsets of selected features were then evaluated independently to determine the optimal combination and arrangement for model development.

Data mining was performed separately for diagnostic and prognostic targets. The final feature sets used in each model are summarized in Table 1.

## Model development

The Random Forest model was selected for its suitability in handling non-parametric medical data. For diagnosis prediction, the dataset was split into training, validation, and test sets in a 3:1:1 ratio. The model was optimized through hyperparameter tuning using grid search, followed by randomized search and stratified cross-validation, ensuring robust performance. To mitigate overfitting and ensure consistent performance across diverse clinical presentations, 10-fold stratified cross-validation was performed on the 555 bootstrap samples of the training set.

**Table 1. List of dharma's features.**

| Diagnosis | Complications |
|---|---|
| Nausea | Nausea |
| Appetite Loss | Appetite Loss |
| Peritonitis | Peritonitis |
| WBC Count | Urinary Ketones |
| Neutrophil Percentage | Free Fluids |
| CRP | CRP |
| Urinary Ketones | WBC Count |
| Appendix Diameter | Body Temperature |
| Free Fluids | Appendix Diameter |
| | Neutrophil Percentage |

The optimal classification threshold was determined using Youden's Index and the Euclidean Distance method on the Receiver Operating Characteristic (ROC) curve applied to the validation set. We aimed to strike a balance between sensitivity and specificity to minimize both false positives and false negatives in appendicitis diagnosis.

The model's performance was evaluated on independent test sets using AUC-ROC, sensitivity, specificity, and predictive values to ensure an unbiased clinical assessment. AUC-ROC was chosen as the primary evaluation metric for the diagnostic task because it captures threshold-independent discriminative performance, allowing decision thresholds to be adjusted post-hoc to suit different clinical subgroups or population cohorts without retraining the entire model.

The 650 diagnosed appendicitis cases, curated as described in the data preprocessing section, were split into training, validation, and test sets in a 3:1:1 ratio. The severity classes were highly imbalanced, with negative-to-positive ratios of 286:104 in the training set, 98:32 in the validation set, and 106:24 in the test set. The higher prevalence of the negative class in the training set biased the model towards predicting uncomplicated cases. To address this, we augmented the training set with 49 complicated cases from the literature [31] and applied NearMiss undersampling to reduce the abundance of easily identifiable uncomplicated cases. This resulted in a more balanced dataset with a ratio of 235:153 (1.54:1).

We further adjusted the class weights by assigning the positive class a weight of 5:1, thereby penalizing false negatives more heavily than false positives. This deliberate design choice biased the prognostic model toward identifying complicated appendicitis and was considered clinically appropriate, as missing complications carries substantially greater risk—such as perforation and gangrene—than incorrectly flagging non-complicated cases. Importantly, all patients in this cohort had a confirmed diagnosis of appendicitis; therefore, a false-positive prediction would typically result in an appendectomy, which is an accepted and established management even for non-complicated appendicitis. In contrast, a false-negative prediction could delay timely surgical intervention in patients with complicated disease, potentially leading to life-threatening outcomes.

The primary evaluation metric for the complication model was recall (sensitivity) for the positive class, reflecting the clinical priority of minimizing missed complications.

We named the final machine-learning framework **Dharma**, which comprises the clinically grounded *Dharma_Imputer* and two Random Forest classifier models: one for diagnosis and the other for prognosis among positively diagnosed cases.

The hyperparameters used for Dharma's Diagnostic and Severity Assessment models are in Table 2.

## Model evaluation

We applied a multi-layer evaluation strategy for Dharma across both diagnostic and prognostic tasks to assess robustness and generalization. First, model performance was estimated using 10-fold stratified cross-validation across 555 bootstrap

**Table 2. Hyperparameters for Dharma.**

| Hyperparameters | Diagnostic Model | Prognostic Model |
|---|---|---|
| n_estimators | 555 | 888 |
| min_samples_split | 12 | 5 |
| min_samples_leaf | 1 | 5 |
| max_depth | 35 | 3 |
| class_weight | "balanced" | {0: 1, 1: 5} |
| max_features | sqrt | sqrt |
| random_state | 17 | 17 |

samples of the training set. Within each bootstrap sample, Dharma was benchmarked against state-of-the-art models appropriate for our dataset size, namely XGBoost and LightGBM.

The diagnostic model was further evaluated on two independent test sets: (i) 5,555 bootstrap samples of the unaltered test split from the primary dataset, and (ii) 5,555 bootstrap samples of 239 unique cases extracted from [21]. For the prognostic (complications) model, the test set contained only 18% complicated cases (106:24). Dharma's severity feature model was benchmarked against XGBoost under two settings: one tuned with simple hyperparameters and another with complex hyperparameters highly regularized for high recall.

Model performance was reported in terms of AUC-ROC, sensitivity, specificity, predictive values, and corresponding 95th-percentile confidence intervals. Hyperparameter tuning for all benchmark models was performed using grid search with stratified 10-fold cross-validation to ensure a fair comparison.

The hyperparameter settings of all benchmark models for diagnostic and prognostic tasks are reported in Tables 3 and 4 respectively.

We further evaluated the robustness of Dharma against different imputation strategies for missing values. For the markers of inflammation (WBC, Neutrophil Percentage, Body temperature, CRP), we applied Iterative Imputer from the Scikit-learn library with decision tree regressor and linear regressor estimators, as well as a K-nearest neighbors (KNN) imputer. For categorical variables, we paired these regression imputation strategies with a decision tree classifier custom-trained for each feature. In addition, we tested simple strategies using mean imputation for continuous variables and mode imputation for categorical variables. Across all methods, no statistically significant difference in model performance was observed, indicating that Dharma's predictive ability was stable irrespective of the imputation approach (S2 Table).

For the final Dharma_Imputer, we selected a combination of decision tree regressor imputation for continuous variables and decision tree classifier imputation for categorical variables. This approach enables real-time, clinically

**Table 3. Hyperparameters of SOTA models for diagnosis.**

| XGBoost | | LightGBM | |
|---|---|---|---|
| n_estimators | 555 | n_estimators | 333 |
| max_depth | 3 | max_depth | 3 |
| min_child_weight | 6 | min_child_samples | 3 |
| learning_rate | 0.01 | learning_rate | 0.01 |
| gamma | 0 | min_gain_to_split | 0 |
| subsample | 0.6 | bagging_fraction | 0.6 |
| colsample_bytree | 0.6 | feature_fraction | 0.6 |
| reg_alpha | 0.1 | lambda_l1 | 0 |
| reg_lamda | 1 | lambda_l2 | 1 |
| random_state | 17 | random_state | 17 |

Table 4. Hyperparameters of SOTA models for complications.

| Hyperparameters | XGB(Simple) | XGB(Complex) |
|---|---|---|
| n_estimators | 111 | 111 |
| max_depth | 3 | 3 |
| min_child_weight | 6 | 6 |
| scale_pos_weight | 7 | 7 |
| learning_rate | 0.01 | 0.01 |
| gamma | – | 1 |
| subsample | – | 0.8 |
| colsample_bytree | – | 0.6 |
| reg_alpha | – | 0.1 |
| reg_lamda | – | 10 |

informed imputation, which is particularly important in real-world settings where missing data are frequent, especially in resource-constrained environments.

## Explainability analysis

To examine Dharma's decision-making process, we applied SHAP's Tree Explainer (SHapley Additive exPlanations), a game theory–based approach that quantifies the contribution of each feature relative to a baseline prediction (base SHAP value). A feature's SHAP value indicates its influence in shifting the predicted probability toward or away from this baseline. Global feature contributions were visualized using SHAP summary plots for both diagnostic and prognostic tasks. Additionally, we employed the Random Forest's native *feature_importance* method to assess the relative importance assigned to each variable by the models.

At the individual level, SHAP values were integrated into our web application alongside 95th-percentile confidence intervals derived from the 555 decision trees in Dharma's random forest, providing real-time uncertainty estimates. This integration enables attending physicians to explore the reasoning behind model predictions, thereby enhancing transparency, interpretability, and clinical trust.

## Results

### Data mining

A total of 780 pediatric patients (ages 0–18 years; mean age: 11.35 years, 95% CI: 11.10–11.59) with suspected appendicitis were included in the diagnostic prediction cohort, with a male-to-female ratio of 1.07 (403:377). For complication prediction, 650 confirmed appendicitis cases from the combined dataset, as described earlier, were included.

A Chi-square test was performed to assess the significance of associations, and Cramér's V was used to evaluate the strength of association of the categorical features. Seven features showed significant association with appendicitis diagnosis and six features were significantly associated with complication prediction ($p < 0.001$). Among the diagnostic features, peritonitis demonstrated the strongest association (Cramér's $V = 0.37$), followed by neutrophilia (Cramér's $V = 0.23$). For complication prediction, peritonitis again showed the strongest association (Cramér's $V = 0.30$), followed by urinary ketones (Cramér's $V = 0.29$), Tables 5 and 6.

A Mann-Whitney U test was conducted to assess the statistical significance of differences in the distributions of continuous variables between positive and negative cases, and rank-biserial correlation was used to evaluate the effect size of these differences, Tables 7 and 8. Six variables were significantly associated with appendicitis diagnosis, and five

**Table 5. Diagnosis and categorical variables (n = 780).**

| Features | Chi-Square | P-value | Cramer's V | Samples |
|---|---|---|---|---|
| Peritonitis | 108.24 | <0.001 | 0.37 | 773 |
| Neutrophilia | 67.17 | <0.001 | 0.30 | 732 |
| Free Fluids | 32.48 | <0.001 | 0.21 | 719 |
| Ketones in Urine | 24.34 | <0.001 | 0.20 | 582 |
| Contralateral Rebound Tenderness | 25.08 | <0.001 | 0.18 | 767 |
| Nausea/Vomiting | 23.19 | <0.001 | 0.17 | 774 |
| Loss of Appetite | 19.24 | <0.001 | 0.16 | 772 |
| Ipsilateral Rebound Tenderness | 8.64 | <0.01 | 0.12 | 619 |
| Sex | 10.78 | <0.01 | 0.12 | 779 |
| RIF Tenderness | 8.18 | <0.01 | 0.10 | 774 |
| Migratory Pain | 6.70 | 0.01 | 0.09 | 773 |
| Coughing Pain | 6.30 | 0.01 | 0.09 | 766 |
| Psoas Sign | 4.00 | 0.05 | 0.07 | 745 |
| Stool | 3.04 | 0.22 | 0.06 | 765 |
| RBC in Urine | 2.02 | 0.57 | 0.06 | 576 |
| WBC in Urine | 1.25 | 0.74 | 0.05 | 583 |
| Dysuria | 1.61 | 0.20 | 0.05 | 753 |

**Table 6. Complications and categorical variables(n = 463).**

| Features | Chi-Square | P-value | Cramer's V | Samples |
|---|---|---|---|---|
| Peritonitis | 41.29 | <0.001 | 0.3 | 457 |
| Ketones in Urine | 26.45 | <0.001 | 0.29 | 318 |
| Neutrophilia | 23.75 | <0.001 | 0.23 | 434 |
| Nausea/Vomiting | 23.17 | <0.001 | 0.22 | 458 |
| Loss of Appetite | 20.56 | <0.001 | 0.21 | 456 |
| Free Fluids | 16.72 | <0.001 | 0.2 | 430 |
| RBC in Urine | 10.65 | 0.01 | 0.18 | 317 |
| Ipsilateral Rebound Tenderness | 5.62 | <0.05 | 0.14 | 305 |
| WBC In Urine | 4.9 | 0.18 | 0.12 | 319 |
| Stool | 5.12 | 0.08 | 0.11 | 451 |
| RIF Tenderness | 2.66 | 0.1 | 0.08 | 458 |
| Psoas Sign | 1.58 | 0.21 | 0.06 | 433 |
| Contralateral Rebound Tenderness | 1.5 | 0.22 | 0.06 | 451 |
| Migratory Pain | 0.59 | 0.44 | 0.04 | 457 |
| Dysuria | 0.54 | 0.46 | 0.04 | 437 |
| Sex | 0.54 | 0.46 | 0.03 | 462 |
| Coughing Pain | 0.01 | 0.94 | 0.004 | 450 |

variables were significantly associated with complication prediction (p < 0.001). Among the diagnostic variables, appendix diameter demonstrated the strongest effect (rank-biserial correlation = 0.9), followed by C-reactive protein (CRP) level (rank-biserial correlation = 0.46). For complication prediction, CRP level exhibited the strongest effect (rank-biserial correlation = 0.64), followed by white blood cell (WBC) count (rank-biserial correlation = 0.45).

**Table 7. Diagnosis and continuous variables (n = 780).**

| Feature | U_statistic | P_value | Rank-Biserial correlation | Mean (Positive Class) | Mean (Negative Class) | Samples |
|---|---|---|---|---|---|---|
| Appendix Diameter | 2341 | <0.001 | 0.9 | 8.7 | 5.04 | 498 |
| CRP | 38686 | <0.001 | 0.46 | 44.9 | 11.72 | 771 |
| WBC Count | 39910 | <0.001 | 0.45 | 14.28 | 10.33 | 776 |
| Neutrophil Percentage | 32518.5 | <0.001 | 0.42 | 76.03 | 65.6 | 679 |
| Body Temperature | 57410 | <0.001 | 0.21 | 37.52 | 37.24 | 775 |
| BMI | 79280 | <0.001 | -0.15 | 18.45 | 19.56 | 753 |
| Weight | 82617 | <0.01 | -0.13 | 41.72 | 45.25 | 778 |
| Age | 80879 | 0.02 | -0.1 | 11.08 | 11.72 | 780 |
| Height | 74276.5 | 0.08 | -0.07 | 146.93 | 149.51 | 755 |
| RDW | 65536.5 | 0.21 | 0.05 | 13.4 | 12.87 | 756 |
| Hemoglobin | 72577.5 | 0.52 | -0.03 | 13.38 | 13.38 | 764 |
| RBC Count | 70347 | 0.92 | 0.004 | 4.79 | 4.82 | 764 |
| Thrombocyte Count | 70459.5 | 0.95 | 0.003 | 285.79 | 284.48 | 764 |

**Table 8. Complications and continuous variables(n = 463).**

| Feature | U_statistic | P_value | Rank-Biserial Correlation | Mean (Positive Class) | Mean (Negative Class) | Samples |
|---|---|---|---|---|---|---|
| CRP | 6952.5 | <0.001 | 0.64 | 107.68 | 24.52 | 457 |
| WBC Count | 10960 | <0.001 | 0.45 | 17.43 | 13.23 | 460 |
| Body Temperature | 12010.5 | <0.001 | 0.4 | 37.93 | 37.38 | 459 |
| Neutrophil Percentage | 8803.5 | <0.001 | 0.38 | 82.44 | 74.14 | 403 |
| Appendix Diameter | 7296 | <0.001 | 0.28 | 9.53 | 8.51 | 371 |
| Weight | 24201.5 | <0.01 | -0.2 | 38.73 | 42.75 | 461 |
| Age | 24327 | <0.01 | -0.2 | 10.08 | 11.42 | 463 |
| Height | 21988.5 | <0.01 | -0.17 | 141.56 | 148.79 | 443 |
| BMI | 21284.5 | 0.01 | -0.16 | 17.92 | 18.64 | 441 |
| Hemoglobin | 21018.5 | 0.06 | -0.12 | 13.36 | 13.39 | 450 |
| Thrombocyte Count | 17298 | 0.2 | 0.08 | 296.65 | 282.24 | 450 |
| RBC Count | 20250.5 | 0.23 | -0.08 | 4.75 | 4.8 | 450 |
| RDW | 17530 | 0.44 | 0.05 | 13.56 | 13.35 | 445 |

## Diagnostic performance

**Conventional tools.** All conventional scoring systems—Alvarado Score (AS), Pediatric Appendicitis Score (PAS), Appendicitis Inflammatory Response (AIR) Score, and Tzanaki Score—were significantly associated with appendicitis diagnosis (p < 0.001). The discriminatory performance, expressed as AUC-ROC with 95% confidence intervals, was 0.76 [0.71–0.82] for AS, 0.70 [0.64–0.76] for PAS, 0.74 [0.68–0.80] for AIR, and 0.91 [0.87–0.94] for the Tzanaki Score. Median values were consistently higher among confirmed cases compared to negatives (AS: 7 vs. 5; PAS: 5 vs. 4; AIR: 6 vs. 4; Tzanaki: 12 vs. 6), indicating higher scoring trends among positive cases, Table 9.

The Pediatric Appendicitis Score (PAS) with a low cut-off of 4 demonstrated the highest sensitivity, 89 [84–93], making it most effective for ruling out the disease. Conversely, the Appendicitis Inflammatory Response (AIR) Score with a cut-off of 9 achieved very high specificity, 99 [97–100], supporting its role in ruling in the disease. A stepwise approach—using PAS (≥ 4) for initial screening and AIR (≥ 9) for final diagnostic confirmation may provide a promising framework for future

**Table 9. Association of scoring systems with diagnosis.**

| Scoring Systems | P-value | Rank-Biserial Correlation | Median (Positive Class) | Median (Negative Class) |
|---|---|---|---|---|
| Alvarado Score | <0.001 | 0.52 | 7 | 5 |
| Pediatric Appendicitis Score | <0.001 | 0.4 | 5 | 4 |
| AIR Score | <0.001 | 0.48 | 6 | 4 |
| Tzanaki Score | <0.001 | 0.82 | 12 | 6 |

studies. However, none of these linear scoring systems, when used in isolation, demonstrated sufficient performance to serve as a reliable standalone diagnostic method, Table 10.

Ultrasonography (USG) demonstrated the highest diagnostic performance among the available tools, with an AUC-ROC of 0.94 [0.89–0.99]. However, appendix diameter measurements were unavailable in 34% of suspected cases, reflecting the inherent challenges in appendix visualization. A similar limitation was observed with the Tzanaki Score, which relies heavily on USG findings for its calculation despite its strong discriminatory ability.

**Dharma.** Dharma demonstrated consistently high diagnostic performance across cross-validation and independent test sets.

In 10-fold cross-validation, the model achieved an AUC-ROC of 0.98 [0.97–0.99] with an overall accuracy of 93% [91–95]. The performance was balanced, with sensitivity 93% [91–95], specificity 93% [90–96], PPV 95% [93–97], and NPV 90% [87–94].

In the first independent test set, the model maintained robust discriminative ability with an AUC-ROC of 0.98 [0.96–0.99] and accuracy of 91% [87–95]. Sensitivity remained high (93% [88–98]) with slightly lower specificity (87% [78–95]), while predictive values were balanced (PPV 93% [88–97], NPV 88% [78–96]).

In the second curated test set, performance was further reinforced, achieving an AUC-ROC of 0.98 [0.96–0.99] and accuracy of 94% [91–97]. The model demonstrated high specificity (98% [95–100]) and PPV (99% [97–100]) while maintaining sensitivity (92% [88–96]) and NPV (87% [80–93]).

Even in the cohort with an unvisualized appendix, Dharma maintained strong discriminatory ability with an AUC-ROC of 0.96 [0.93–0.99]. At a threshold of 44%, it achieved a very high specificity of 97% [93–100] with a PPV of 93% [84–100]. At a threshold of 25%, it achieved a high NPV of 95% [91–99] and sensitivity of 92% [84–98].

Overall, Dharma showed excellent and stable diagnostic accuracy with consistently high AUC (>0.96) across all evaluations, demonstrating both generalizability and clinical robustness. Detailed diagnostic results are summarized in Table 11 and Table 12.

**Table 10. Performance of diagnostic tools on curated test set (n=289).**

| Tools | AUC-ROC | Sensitivity % | Specificity % | PPV % | NPV % | Accuracy % |
|---|---|---|---|---|---|---|
| Alvarado Score(≥5) | 0.76 [0.71-0.82] | 86 [81-91] | 45 [35-55] | 68 [74-80] | 64 [53-75] | 72 [66-78] |
| Alvarado Score(≥7) | 0.76 [0.71-0.82] | 55 [48 –62] | 83[76-90] | 86 [79-92] | 50 [43 –58] | 65 [60 –70] |
| PAS Score(≥4) | 0.70 [0.64-0.76] | 89 [84-93] | 38 [29-48] | 72 [67-78] | 65 [53-76] | 71 [66-76] |
| PAS Score(≥6) | 0.70 [0.64-0.76] | 48 [40 –55] | 76 [68-85] | 79 [71-86] | 44 [3] | 58 [52 –63] |
| AIR Score(≥5) | 0.74 [68-80] | 67 [60 –73] | 67 [57-76] | 79 [72-85] | 52 [44-61] | 67 [61 –72] |
| AIR Score(≥9) | 0.74 [68-80] | 16 [11 –21 ] | 99 [97-100] | 97 [89-100] | 39 [33 –45] | 45 [40 –51 ] |
| Tzanaki Score(≥8) | 0.91 [87-94] | 85 [80-90] | 88 [81-94] | 93 [89-97] | 76 [68-84] | 86 [82-90] |
| USG(≥6mm) | 0.94 [0.89-0.99] | 94 [91-97] | 95 [87-100] | 99 [97-100] | 80 [67-91] | 94 [91-97] |

**Table 11. Dharma's diagnostic performance.**

| Dharma | AUC-ROC | Specificity | PPV | Sensitivity | NPV | Accuracy |
|---|---|---|---|---|---|---|
| 10-fold cross- validation | 0.98 [0.97-0.99] | 93% [90-96] | 95% [93-97] | 93% [90-95] | 90% [87-94] | 93% [91-95] |
| Split Test Set(n = 156) | 0.98 [0.96-0.99] | 87% [78-95] | 93% [88-97] | 93% [88-98] | 88% [78-96] | 91% [87-95] |
| Curated Test Set(n = 289) | 0.98 [0.96-0.99] | 98% [95-100] | 99% [97-100] | 92% [88-96] | 87% [80-93] | 94% [91-97] |

**Table 12. Diagnostic performance on unvisualized appendix cohort (n = 144).**

| Threshold | AUC-ROC | Specificity | PPV | Sensitivity | NPV | Accuracy |
|---|---|---|---|---|---|---|
| 44% | 0.96 [0.93-0.99] | 97% [93-100] | 93% [84-100] | 75% [62-86] | 87% [80-93] | 89% [83-94] |
| 25% | 0.96 [0.93-0.99] | 89% [83-95] | 82% [72-92] | 92% [84-98] | 95% [91-99] | 90%[85-95] |

## Prognostic performance

For complication prediction, Dharma achieved a sensitivity of 96% [93–99] and NPV of 97% [94–99] in 10-fold cross-validation, with corresponding specificity 65% [53–75], PPV 65% [59 –72], and AUC-ROC 0.92 [0.90–0.95]. This performance profile, characterized by high sensitivity and NPV, reflects the intended design of Dharma's severity assessment module as a screening tool to safely rule out complications and to guide decisions regarding surgical versus conservative management, as well as the urgency of surgery.

In the unaltered independent test set, the model again demonstrated strong recall for the positive class, with sensitivity 88% [68–97] and NPV 95% [85–99]. Specificity was lower (49% [39–59]) with a corresponding PPV of 28% [18–40]. Confidence intervals for this evaluation were calculated using Clopper–Pearson exact binomial estimation, which is appropriate for small sample sizes and rare events. This was necessary given the pronounced class imbalance (106 negative vs. 24 positive cases). The wide confidence interval observed for sensitivity likely reflects the limited number of positive cases (24, 18% of the test set).

Overall, Dharma demonstrated strong prognostic performance for complication prediction, with high sensitivity and NPV supporting its role as a reliable screening tool. Detailed prognostic results are summarized in Table 13.

## Benchmarking analyses

**Diagnostic value.** We compared Dharma, the Imputer–Diagnostic Classifier, against existing clinical scoring systems, ultrasonography (USG), and two state-of-the-art models suited for medium-sized tabular data (XGBoost and LightGBM). Evaluation was performed using 5,555 bootstrap samples of the curated test set (n = 289) and the split test set.

Dharma consistently outperformed all existing scoring systems across every key metric (Table 14). Against SOTA models, Dharma was slightly inferior in the split test set but showed a slight edge in the curated test set (Table 15). Taken together, these findings suggest Dharma performs on par with contemporary machine-learning benchmarks while clearly surpassing conventional diagnostic tools.

**Table 13. Dharma's performance for severity assessment.**

| Dharma | Sensitivity | NPV | Specificity | PPV |
|---|---|---|---|---|
| 10-fold cross-validation | 96% [93-99] | 97% [94-99] | 65% [53-75] | 65% [59 –72] |
| Unaltered Test Set (n = 130) | 88% [68-97] | 95% [85-99] | 49% [39-59] | 28% [18-40] |

**Table 14. Dharma minus conventional tools on curated test set (n = 289).**

| Tool | AUC-ROC | Accuracy | Sensitivity | Specificity | PPV | NPV |
|------|---------|----------|-------------|-------------|-----|-----|
| AS (≥5) | 0.22 [0.16–0.27] | 0.23 [0.17–0.28] | 0.06 [0.01–0.11] | 0.53 [0.42–0.63] | 0.25 [0.19–0.31] | 0.23 [0.12–0.34] |
| AS (≥7) | 0.22 [0.16–0.27] | 0.29 [0.24–0.35] | 0.37 [0.30–0.44] | 0.15 [0.07–0.23] | 0.13 [0.07–0.20] | 0.37 [0.30–0.44] |
| PAS (≥4) | 0.28 [0.22–0.34] | 0.23 [0.17–0.29] | 0.03 [−0.03–0.09] | 0.60 [0.49–0.70] | 0.26 [0.21–0.32] | 0.22 [0.10–0.34] |
| PAS (≥6) | 0.28 [0.22–0.34] | 0.36 [0.30–0.42] | 0.44 [0.36–0.52] | 0.22 [0.13–0.31] | 0.20 [0.13–0.28] | 0.43 [0.35–0.50] |
| AIR (≥5) | 0.24 [0.18–0.30] | 0.27 [0.21–0.33] | 0.25 [0.18–0.32] | 0.31 [0.22–0.41] | 0.20 [0.14–0.27] | 0.35 [0.27–0.43] |
| AIR (≥9) | 0.24 [0.18–0.30] | 0.49 [0.43–0.55] | 0.76 [0.69–0.82] | −0.01 [−0.04–0.02] | 0.02 [−0.02–0.10] | 0.48 [0.41–0.54] |
| Tzanaki | 0.07 [0.04–0.10] | 0.08 [0.04–0.12] | 0.07 [0.03–0.11] | 0.10 [0.04–0.16] | 0.06 [0.03–0.10] | 0.11 [0.05–0.17] |
| USG | 0.04 [0.00–0.09] | 0.01 [0.00–0.03] | 0.01 [0.00–0.03] | 0.00 [0.00–0.00] | 0.00 [0.00–0.00] | 0.04 [0.00–0.10] |

**Table 15. Dharma minus SOTA models for diagnostic task in two test sets.**

| Metrics | Split test set (n=156) | | Curated test set (n=289) | |
|---------|------------------------|---|--------------------------|---|
| | XGBoost | LightGBM | XGBoost | LightGBM |
| AUC-ROC | 0.00[-0.01-0.00] | 0.00[-0.01-0.00] | 0.01[0.00-0.02] | 0.01[0.00-0.01] |
| Accuracy | -0.02[-0.04-0] | -0.01[-0.03-0.01] | 0.00[-0.01-0.01] | 0.00[-0.01-0.01] |
| Sensitivity | -0.03[-0.07-0] | -0.01[-0.04-0.02] | -0.01[-0.02-0.01] | -0.01[-0.02-0.-01] |
| Specificity | 0.00[0.00-0.00] | 0.00[0.00-0.00] | 0.01[0.00-0.03] | 0.01[0-0.03] |
| PPV | 0.00[-0.01-0] | 0.00[0.00-0.00] | 0.01[0.00-0.02] | 0.01[0-0.02] |
| NPV | -0.05[-0.11-0] | -0.02[-0.07-0.04] | -0.01[-0.03-0.02] | -0.01[-0.03-0.02] |

## Prognostic performance

Dharma was marginally inferior to the XGBoost models in key prognostic metrics, namely sensitivity and NPV, but outperformed them markedly in secondary metrics, including accuracy, specificity, and PPV. A detailed comparison is provided in Table 16.

## SHAP analysis

SHAP (Shapley Additive Explanations) analysis was used to interpret the Dharma model's predictions. For the diagnostic model, the SHAP base value—representing the expected model output prior to incorporating feature-specific contributions—was 0.50. In contrast, the severity assessment model exhibited a higher base value of 0.73. This difference reflects an intentional and clinically motivated design choice: the diagnostic model was optimized for balanced discrimination, whereas the severity model, applied exclusively to confirmed appendicitis cases, was deliberately biased toward the

**Table 16. Dharma minus SOTA models for prognostic task.**

| Metric | XGBoost (Simple Hyperparameters) | XGBoost (Complex Hyperparameters) |
|--------|----------------------------------|-----------------------------------|
| Sensitivity | −0.01 [−0.05–0.01] | −0.03 [−0.07–0.01] |
| NPV | −0.01 [−0.04–0.02] | −0.01 [−0.05–0.10] |
| Specificity | 0.18 [0.08–0.27] | 0.40 [0.29–0.51] |
| AUC-ROC | 0.01 [−0.01–0.02] | 0.00 [−0.01–0.01] |
| Accuracy | 0.10 [0.04–0.15] | 0.23 [0.17–0.29] |
| PPV | 0.10 [0.05–0.14] | 0.18 [0.14–0.23] |

positive (complicated) class to minimize the risk of missing potentially complicated disease (false negatives), which carries substantial clinical consequences if surgical intervention is delayed.

In the diagnostic task, appendix diameter exerted the greatest influence on predictions, followed by white blood cell (WBC) count, C-reactive protein (CRP), and neutrophil percentage. For the complications model, CRP had the strongest impact, followed by appendix diameter and peritonitis. Interestingly, larger appendix diameters were associated with lower predicted complication risk, suggesting an inverse relationship between diameter and severity that warrants further exploration.

The absence of appendix diameter data reduced the model's diagnostic probability below the base value; subsequent contributions from other features then shifted the prediction upward to reach the final output. Under these circumstances, peritonitis had the strongest influence, followed by CRP, WBC count, and neutrophil percentage. The global feature importance patterns for both diagnostic and prognostic tasks are illustrated in the SHAP summary dot plots (S1–S3 Figs).

## Discussion

In this study, we present *Dharma*, a novel and clinically grounded machine learning framework specifically designed for pediatric appendicitis. Dharma integrates its native imputer (*Dharma_Imputer*) with a stacked architecture comprising two Random Forest models.

The first model, optimized for unbiased diagnostic performance, achieved an AUC–ROC of 0.98 [0.97–0.99] and an accuracy of 93% [91–95] in identifying acute appendicitis during 10-fold cross-validation on 555 bootstrap samples of the training set. Model performance was well balanced, with a sensitivity of 93% [91–95], specificity of 93% [90–96], positive predictive value (PPV) of 95% [93–97], negative predictive value (NPV) of 90% [87–94], and a base SHAP value of 0.50. This balanced performance was achieved through the use of a relatively balanced training dataset and hyperparameter tuning with AUC–ROC as the primary optimization metric, and using class weight as balanced. These design choices reflect the clinical implications of misclassification in appendicitis diagnosis: false negatives may increase morbidity by delaying treatment and predisposing patients to complications, whereas false positives may lead to unnecessary interventions, including negative appendectomy or advanced but complex imaging in pediatric cohorts.

For patients classified as having acute appendicitis by the diagnostic model, a second Random Forest model was deployed for prognostic assessment. This severity model was intentionally optimized to prioritize detection of complicated appendicitis, achieving a sensitivity of 96% [93–99] and a negative predictive value (NPV) of 97% [94–99], with a specificity of 65% [53–75], positive predictive value (PPV) of 65% [59 –72], and an AUC–ROC of 0.92 [0.90–0.95] based on cross-validation. The base SHAP value of 0.73 reflects this deliberate operating point favoring the positive (complicated) class. For the severity assessment task, Dharma was designed to minimize false negatives, as delayed recognition of complicated appendicitis carries substantial clinical risk. In contrast, false-positive predictions in this setting primarily lead to appendectomy, which remains the standard management even for uncomplicated appendicitis. This clinically motivated optimization was achieved through augmentation of complicated cases from the existing literature, selection of recall (sensitivity) for the positive class as the primary hyperparameter optimization metric, and assignment of a 5:1 class weight favoring the positive class.

Through this approach, Dharma provides a highly reliable and well-calibrated diagnostic tool for a common yet challenging pediatric emergency, while also functioning as a risk-aware screening tool for potential complications, supporting safe exclusion and early, evidence-based management decisions.

To assess its performance, Dharma was compared with established linear scoring systems, including both older models (Alvarado Score, Pediatric Appendicitis Score [PAS]) and newer models (Appendicitis Inflammatory Response [AIR] Score, Tzanaki Score), on a common, previously unseen test cohort (n = 289). Each scoring system showed strengths and weaknesses: for instance, AIR with a high cut-off of 9 demonstrated excellent specificity for appendicitis diagnosis (99

[97–100]), whereas PAS with a low cut-off of 4 was highly sensitive (89 [84–93]). Using PAS (≤4) for ruling out and AIR (≥9) for ruling in may provide a simple yet pragmatic framework for clinical decision-making, which warrants investigation in future studies.

However, none of these scoring systems alone is sufficiently robust for diagnosing acute appendicitis. Their commonly used intermediate cut-offs have relatively poor specificity (38%–67%), leading to a high rate of false positives and subsequent negative appendectomies. Similar patterns were noted in [32], who reported specificities as low as (14.3%–57.1%). These limitations underscore the need for more accurate, balanced and data-driven non-linear approaches such as Dharma.

As expected, ultrasonography (USG) demonstrated excellent diagnostic performance in our test cohort (n = 289), with an AUC-ROC of 0.94 [0.89–0.99]. Using a 6 mm appendix diameter cutoff, sensitivity and specificity reached 95% [87–100] and 95% [87–100], respectively. However, in 34% of cases the appendix could not be visualized. This limitation aligns with prior reports [33,34], that described non-visualization rates as high as 71–76% in suspected appendicitis. Failure to identify the appendix may be attributed to anatomical variation, patient habitus, or the operator-dependent nature of USG [35]. Moreover, advanced features beyond diameter—such as peri-appendiceal fluid collection (PALC), fat stranding, target sign, hyperemia on Color Doppler, or free fluid—are typically recognized only by experienced radiologists, who are usually not available in primary care or resource-limited contexts. Importantly, USG findings must be integrated with clinical and laboratory parameters for accurate diagnosis [36], and they provide little prognostic information about disease severity. In LMICs and overcrowded emergency departments, even timely USG access may be limited, further constraining its practical utility.

Advanced imaging modalities such as CT and MRI provide high diagnostic accuracy for appendicitis [37–39,40]. However, their use in children is limited by the risks of sedation, contrast exposure, and ionizing radiation [41]. In addition, these modalities remain largely inaccessible in many primary and secondary healthcare settings, particularly in resource-limited regions like ours. Crucially, while they improve diagnostic certainty, they do not reliably differentiate between uncomplicated and complicated appendicitis—an essential distinction for guiding timely and appropriate management [42].

Significant progress has been made in applying machine learning to the diagnosis, severity stratification, and management of pediatric appendicitis [21,20,22–26]. Yet, many of these studies are highly technical and AI-centric, which makes their predictions difficult for clinicians to interpret and incorporate into day-to-day practice. Unlike traditional bedside scoring systems, machine learning models also require dedicated, user-friendly interfaces to be clinically useful. For example, the image-based approach [24], although innovative, is computationally intensive and impractical for real-time decision-making in busy emergency departments. Similarly, the online tool developed by [21] improved accessibility but remains dependent on complex ultrasonographic variables requiring expert radiological input. Furthermore, several features included in their dataset were not directly used in the model's decision process, which diminishes transparency and clinical applicability. As a result, such tools are better suited to academic exploration and dataset development rather than to frontline clinical deployment.

Thus, to address the common yet challenging clinical problem of diagnosing and grading acute appendicitis, we present **Dharma**, a novel non-linear, multimodal framework deployed as a real-time clinical decision-support web application (dharma-ai.org). Dharma first performs robust imputation of features frequently missing in resource-constrained settings—such as C-reactive protein (CRP), appendiceal diameter, free fluid, and urinary ketones—using the **Dharma Imputer**, a model pre-trained on the appendicitis cohort. The complete feature set is then passed to the downstream, class-balanced diagnostic model, which outputs a probability score (the **Dharma score**) indicating the likelihood of acute appendicitis. For cases predicted as positive, a second downstream prognostic model—optimized for high recall for the positive (complicated) class—stratifies the case as *complicated* or *non-complicated*. Both model decisions are accompanied by corresponding SHAP-based explanations to enhance interpretability and clinical transparency.

Importantly, Dharma not only substantially surpasses traditional linear scoring systems—as demonstrated in our benchmarking analyses—but also outperforms all previously published machine-learning models for both diagnostic classification and severity prediction of appendicitis [20–26,29,43]. Beyond these performance gains, Dharma has been translated into a fully deployed, real-time clinical decision support system accessible through a functional web interface. We consider this practical implementation to be a more meaningful contribution than incremental improvements in model metrics alone, as it directly addresses the translational gap that limits most existing research models.

When benchmarked against contemporary state-of-the-art machine learning architectures such as XGBoost and LightGBM, which are designed for handling incomplete non-parametric tabular datasets, Dharma demonstrated broadly comparable performance across both diagnostic and prognostic tasks on unseen test sets. Although some differences reached statistical significance, they were clinically negligible, indicating that Dharma provides comparable discriminative performance while offering distinct advantages in simplicity, accessibility, interpretability, and clinical alignment.

Dharma assesses suspected cases of acute appendicitis by analyzing a comprehensive non-linear pattern of clinical features. These include symptoms such as nausea or vomiting and loss of appetite; signs like peritonitis; physiological parameters such as body temperature; laboratory findings including white blood cell (WBC) count, percentage of neutrophils, C-reactive protein (CRP), and urinary ketones; as well as radiological findings such as appendiceal diameter and the presence of free fluid on ultrasonography. These features are already well-established in existing literature for the diagnosis and risk stratification of acute appendicitis [44–49].

While the assessment of signs and symptoms can be particularly challenging in younger children, certain features remain comparatively more accessible. History-based features such as nausea/vomiting and poor feeding are generally easier to elicit, even in preverbal children through caregiver observations. Similarly, signs of peritonitis can often be assessed using surrogate physical findings, including tenderness, rebound tenderness, guarding, and rigidity—either localized to the right iliac fossa (RIF) or generalized across the abdomen. Radiological parameters such as appendiceal diameter and the presence of peritoneal free fluid are also reliably assessable in most pediatric patients, even with screening ultrasonography or Point-Of-Care Ultrasound (POCUS), making them particularly valuable in settings where advanced imaging modalities or experienced radiologists may be unavailable or limited.

Dharma is a medicine-first, hybrid machine learning framework designed by integrating clinical reasoning into every aspect of its development. From clinically informed feature selection and data imputation to an unbiased diagnostic model, a risk-averse severity assessment model, and deployment through a fully functional web application, Dharma follows a structured, stepwise approach. This workflow demonstrates that clinical reasoning can be directly encoded into models and systems through careful, medicine-first architectural design, increasing the likelihood of developing models that generalize effectively, provide explainable outputs, and operate within real-world clinical constraints [16,17].

A key contribution of this work is the translation of Dharma into a practical, transparent, and interpretable web-based tool. The platform provides Dharma Score, case-level probability estimate with 95% confidence intervals derived from the 555 estimators in the diagnostic random forest, as well as the predicted probability of chances of complications, thereby supporting surgeons in selecting appropriate treatment modalities. To enhance interpretability, Dharma also generates feature-level SHAP explanations for both diagnostic and prognostic tasks, allowing clinicians to understand and trust the model's reasoning. For real-time deployment, the Dharma_Imputer has been integrated into the web application pipeline to intelligently impute variables such as CRP, appendix diameter, urinary ketones and free fluid, which are often difficult to assess in resource-limited settings. Other variables have been made mandatory, as they are relatively easier to obtain even under such constraints.

Thus, Dharma is not merely another high-performing algorithm but a purpose-built, clinically grounded system designed for real-world, bedside use, S4 Fig. More broadly, we suggest that future progress in clinical AI may hinge less on incremental algorithmic optimization and more on the development of systems that are clinically aligned, resource-sensitive, interpretable, and usable.

We believe that knowledge should be freely available and openly shared. In alignment with this principle, we have made our source code publicly accessible on GitHub [50], and the web application is freely accessible at dharma-ai.org, promoting transparency, reproducibility, and continuous improvement in medical science. Screenshots of the interface are provided in S4–S6 Figs.

Nevertheless, this study has certain limitations that warrant acknowledgment. Foremost, the lack of access to large, high-quality medical datasets remains a significant barrier, limiting the full potential of statistical and machine learning approaches that we believe are essential for advancing 21st-century medicine [51]. We strongly advocate for the open sharing of anonymized clinical datasets to accelerate broader and more impactful research. Another limitation is that our model was developed on single-center data, raising questions about its adaptability across diverse populations. Furthermore, the absence of exclusively histologically confirmed appendicitis cases may have constrained the model's ability to fully optimize pattern recognition in both diagnostic and prognostic tasks. Future work should prioritize multicenter validation—both prospective and retrospective—including studies in low- and middle-income countries (LMICs) to rigorously assess Dharma's real-world clinical utility and generalizability. We also encourage independent researchers and med-tech innovators to validate Dharma further, and where necessary, calibrate its thresholds to suit the epidemiological and clinical nuances of their own populations.

Despite these limitations, the Dharma represents more than an incremental step in clinical decision support. By integrating readily available clinical, laboratory, and radiological findings into a unified predictive framework, it provides real-time, evidence-based diagnostic and prognostic support for one of the most common pediatric surgical emergencies. With its high specificity and positive predictive value, Dharma has the potential to meaningfully reduce unnecessary negative appendectomies while maintaining patient safety through very low false-negative rates. In doing so, the model could lessen reliance on advanced imaging such as CT or MRI, thereby reducing the risks associated with sedation, contrast exposure, and ionizing radiation in children—offering both clinical and economic benefits.

Importantly, Dharma is designed to function across diverse healthcare settings, from resource-limited facilities to advanced tertiary centers. In primary and secondary care, where even basic imaging and surgical expertise may be unavailable, Dharma leverages accessible clinical and laboratory features to support early recognition, guide timely referrals, and reduce delays in management. In high-volume tertiary emergency departments, it may help prioritize cases when imaging is inconclusive or radiology services are saturated. At advanced surgical centers, calibrated thresholds optimized for high specificity and PPV allow Dharma to be used as a reliable rule-in tool. Combined with its severity assessment feature, it can also assist in triaging patients for appendectomy and informing decisions between conservative versus surgical management [52].

Taken together, Dharma illustrates how data-driven tools can complement, rather than replace, clinical judgment. By bridging bedside intuition with machine learning–derived insights, it provides a pathway toward equitable, AI-augmented care. While prospective multicenter validation remains essential, this work underscores the potential of interpretable, accessible, medicine-first, AI-powered, clinical decision support systems—not only for pediatric appendicitis but also as a clinically grounded framework adaptable to broader healthcare challenges.

## Supporting information

**S1 Fig. SHAP summary plot for Dharma's diagnostic model.**
(TIF)

**S2 Fig. SHAP summary plot for Dharma's diagnostic model in patients with missing appendix diameter data.**
(TIF)

**S3 Fig. SHAP summary plot for Dharma's prognostic model.**
(TIF)

**S4 Fig. Dharma as a Clinical Decision Support system (CDSS).**
(TIF)

**S5 Fig. User interface of Dharma's web-based clinical decision support tool. Accessible through dharma-ai.org .**
(TIF)

**S6 Fig. Example of diagnostic and prognostic predictions generated by Dharma. Dharma score and severity risk estimation, combined into a decision framework.**
(TIF)

**S7 Fig. Feature-level SHAP explanations for both the diagnostic and prognostic estimations.**
(TIF)

**S1 Table. Percentage of missing values in each feature.**
(XLSX)

**S2 Table. Performance stability of Dharma with different imputation strategies.**
(XLSX)

## Acknowledgments

We sincerely thank our dear friend, Amrit Neupane, for his continued support and invaluable contributions to the UI/UX and logo design of Dharma. We also extend our gratitude to Marcinkevics and team for making their anonymized datasets publicly available, which greatly facilitated this work.

## Author contributions

**Conceptualization:** Anup Thapa Kshetri, Subash Pahari.

**Data curation:** Anup Thapa Kshetri, Subash Pahari.

**Formal analysis:** Anup Thapa Kshetri.

**Investigation:** Anup Thapa Kshetri.

**Methodology:** Anup Thapa Kshetri.

**Project administration:** Anup Thapa Kshetri.

**Resources:** Anup Thapa Kshetri.

**Software:** Anup Thapa Kshetri, Subash Pahari.

**Supervision:** Anup Thapa Kshetri.

**Validation:** Anup Thapa Kshetri.

**Visualization:** Anup Thapa Kshetri.

**Writing – original draft:** Anup Thapa Kshetri, Shashank Timilsina, Binay Chapagain.

**Writing – review & editing:** Anup Thapa Kshetri.

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
