## [Decision Letter · Decision Letter 0]

29 Jul 2025

Response to Reviewers
Revised Manuscript with Track Changes
Manuscript
**Journal Requirements:**

1. Please insert an Ethics Statement at the beginning of your Methods section, under a subheading 'Ethics Statement'.

2. We note that your Data Availability Statement is currently as follows: “The dataset has been anonymized and is publicly available for secondary use [44].”

**Additional Editor Comments (if provided):**
**Reviewers' Comments:**

**Comments to the Author**

1. Does this manuscript meet PLOS Digital Health’s publication criteria?

Reviewer #1: No

Reviewer #2: Yes

2. Has the statistical analysis been performed appropriately and rigorously?

Reviewer #1: Yes

Reviewer #2: No

3. Have the authors made all data underlying the findings in their manuscript fully available (please refer to the Data Availability Statement at the start of the manuscript PDF file)?

Reviewer #1: Yes

Reviewer #2: Yes

4. Is the manuscript presented in an intelligible fashion and written in standard English?

Reviewer #1: Yes

Reviewer #2: Yes

Reviewer #1: I have gone through the manuscript titled "Dharma: A novel machine learning framework for pediatric appendicitis—diagnosis, severity assessment and evidence-based clinical decision support.".

I have the following observations regarding the manuscript:

1.Inadequate data source and representativeness: The main data were from the Children's Hospital of Regensburg, Germany. Although literature data were supplemented, the lack of validation in multi-center and different regions may affect the universality of the model in resource-constrained areas.

2.Supplement systematic comparisons with existing ML tools to clarify technical advantages: Only the Random Forest model was used without systematic comparisons with other mainstream ML algorithms (such as GBM, XGBoost, CatBoost), making it difficult to verify the universality of the model's superiority.

3.No images provided.

4.Format specifications: The titles of Table 3 and Table 4 are separated from the tables. Some literature citations lack academic DOI numbers (such as [8], [35], [44], [45], [46]), and some literature citations lack page ranges.

Based on these observations, I would like to give a 'Reject' of the manuscript as decision. All the best to the authors!

Regards

Reviewer #2: Dear Authors,

Here are my comments:

1) The paper would benefit if you report your CV metric results along with 95% CI, which gives better readibility than SD.

2) The tables need some tidying, for eg in Table 9. why do you ommit reporting for specificity, NPV, PPV in the 10-fold CV results? These should be added since only the CV results are what best predicits the real world model perfomrance

3) In the missing value imputation strategy it is stated that null values are imputed with zeros, is this only for categoriacal variables or all variables? Because, zero is an actual and clinically meaningful value for several variables (e.g., CRP = 0 mg/L). I would be better to replace with an evidence-based imputation method (multiple imputation or model-specific missingness indicator). Report per-variable missingness and confirm performance stability.

4) I think you should also address the limits and trade-offs of using SMOTE in the study's limitations section

5) You only train a random forest model, the paper would benefit if you also train and compare it with other SOTA tabular data models, eg Xgboost or LightGBM + maybe also a deeplearning based model like TabNet

**Do you want your identity to be public for this peer review?** For information about this choice, including consent withdrawal, please see our Privacy Policy

Reviewer #1: No

Reviewer #2: No

**Figure resubmission:****Reproducibility:** To enhance the reproducibility of your results, we recommend that authors of applicable studies deposit laboratory protocols in protocols.io, where a protocol can be assigned its own identifier (DOI) such that it can be cited independently in the future. Additionally, PLOS ONE offers an option to publish peer-reviewed clinical study protocols. Read more information on sharing protocols at https://plos.org/protocols?utm_medium=editorial-email&utm_source=authorletters&utm_campaign=protocols

---

## [Decision Letter · Decision Letter 1]

27 Nov 2025

Response to Reviewers
Revised Manuscript with Track Changes
Manuscript
**Journal Requirements:**
**Additional Editor Comments (if provided):**
**Reviewers' Comments:**

**Comments to the Author**

Reviewer #2: All comments have been addressed

Reviewer #3: All comments have been addressed

publication criteria?

Reviewer #2: Partly

Reviewer #3: Yes

3. Has the statistical analysis been performed appropriately and rigorously?

Reviewer #2: Yes

Reviewer #3: Yes

4. Have the authors made all data underlying the findings in their manuscript fully available (please refer to the Data Availability Statement at the start of the manuscript PDF file)?

Reviewer #2: Yes

Reviewer #3: Yes

5. Is the manuscript presented in an intelligible fashion and written in standard English?

Reviewer #2: Yes

Reviewer #3: Yes

Reviewer #2: The authors have addressed all my comments in a satisfactory manner.

Reviewer #3: This manuscript presents a novel machine learning–based framework, “Dharma,” for diagnosing and grading pediatric appendicitis, with potential value for evidence-based clinical decision support. The topic is highly relevant to digital health. Some minor improvements would add up better before publication.

Methodology section

1. Any inclusion/exclusion criteria mentioned for inclusion of sample in methods section.

2. Basis for the sample size selection and sampling methods

Things to be considered in discussion section

3. Other AI or machine learning models than has been researched for the appendicitis, if there is any comparison of your findings with them in discussion sections

4. Any superiority this model or the study findings has on Alvarado scale or other any scales used in the clinical diagnosis of appendicitis.

A promising digital health contribution, but significant revisions are required to ensure methodological robustness, transparency, and clinical applicability.

**Do you want your identity to be public for this peer review?** For information about this choice, including consent withdrawal, please see our Privacy Policy

Reviewer #2: **Yes:**  Josip Vrdoljak

Reviewer #3: No

**Figure resubmission:**

**Reproducibility:** To enhance the reproducibility of your results, we recommend that authors of applicable studies deposit laboratory protocols in protocols.io, where a protocol can be assigned its own identifier (DOI) such that it can be cited independently in the future. Additionally, PLOS ONE offers an option to publish peer-reviewed clinical study protocols. Read more information on sharing protocols at https://plos.org/protocols?utm_medium=editorial-email&utm_source=authorletters&utm_campaign=protocols

---

## [Decision Letter · Decision Letter 2]

26 Dec 2025

Response to Reviewers
Revised Manuscript with Track Changes
Manuscript
**Journal Requirements:**
**Additional Editor Comments (if provided):**
**Reviewers' Comments:**

**Comments to the Author**

Reviewer #3: All comments have been addressed

Reviewer #4: All comments have been addressed

publication criteria?

Reviewer #3: Yes

Reviewer #4: Yes

3. Has the statistical analysis been performed appropriately and rigorously?

Reviewer #3: Yes

Reviewer #4: Yes

4. Have the authors made all data underlying the findings in their manuscript fully available (please refer to the Data Availability Statement at the start of the manuscript PDF file)?

Reviewer #3: Yes

Reviewer #4: Yes

5. Is the manuscript presented in an intelligible fashion and written in standard English?

Reviewer #3: Yes

Reviewer #4: Yes

Reviewer #3: The comments has been addressed adequately.

Reviewer #4: 1. The Base SHAP value of 0.73 for the severity model indicates a strong focus on complicated cases. The choice to set the class weights at 5:1 aims to reduce missed complications, which are serious. It is important to highlight this in the Discussion section to avoid misunderstanding of the term "bias."

2. This manuscript compares its results to state-of-the-art models, including XGBoost and LightGBM. However, it could better explain these comparisons in the context of how artificial intelligence and machine learning are being used in medical diagnostics and prognostics. Referring to recent literature would strengthen this section.

3. In the Discussion section, it is necessary to state clearly that the high base value (0.73) and the 5:1 class weight were intentional strategies to improve recall (sensitivity) and NPV for complicated cases. Missing complications is much riskier than a false positive result in a known appendicitis case. Therefore, the way "bias" is described in the Discussion (lines 478-479) should reflect this important clinical focus.

4. It is suggested to refer from the paper: “COVID-19 Pandemic: A Worldwide Critical Review with the Machine Learning Model-Based Prediction”. This source provides a comprehensive overview of AI and machine learning in disease diagnosis and prediction, making it valuable for the Introduction and Discussion sections to illustrate the broad applications of machine learning. It is important to clarify or standardize the reported metrics since metrics from cross-validation offer a more reliable basis for strong claims.

5. It would be beneficial to include recommended high-quality references on AI and machine learning in healthcare (e.g., Rank 1 and Rank 2 references) to better showcase Dharma's strengths, such as hybrid models, optimization, and interpretability, within a stronger academic context.

6. Ensure that all content and reference section , especially in table footnotes and headers, follow the referencing style required by the target journal (e.g., using APA as a standard unless specified otherwise).

7. Change this line to reflect cross-validation results: "This prognostic model focused on achieving recall for complicated cases, reaching a sensitivity of 96% [93–99] and a negative predictive value (NPV) of 97% [94–99] based on 10-fold cross-validation results."

**Do you want your identity to be public for this peer review?** For information about this choice, including consent withdrawal, please see our Privacy Policy

Reviewer #3: **Yes:**  Dr. Purushottam Adhikari

Reviewer #4: No

**Figure resubmission:**

**Reproducibility:** To enhance the reproducibility of your results, we recommend that authors of applicable studies deposit laboratory protocols in protocols.io, where a protocol can be assigned its own identifier (DOI) such that it can be cited independently in the future. Additionally, PLOS ONE offers an option to publish peer-reviewed clinical study protocols. Read more information on sharing protocols at https://plos.org/protocols?utm_medium=editorial-email&utm_source=authorletters&utm_campaign=protocols

---

## [Decision Letter · Decision Letter 3]

7 Jan 2026

Dharma: A novel, clinically grounded machine learning framework for pediatric appendicitis—diagnosis, severity assessment and evidence-based clinical decision support.

PDIG-D-25-00316R3

Dear Dr. Thapa Kshetri,

We are pleased to inform you that your manuscript 'Dharma: A novel, clinically grounded machine learning framework for pediatric appendicitis—diagnosis, severity assessment and evidence-based clinical decision support.' has been provisionally accepted for publication in PLOS Digital Health.

Best regards,

Hadi Ghasemi

Academic Editor

PLOS Digital Health

**Additional Editor Comments (if provided):**

**Reviewer Comments (if any, and for reference):**

Reviewer's Responses to Questions

**Comments to the Author**

Reviewer #4: All comments have been addressed

publication criteria?

Reviewer #4: Yes

3. Has the statistical analysis been performed appropriately and rigorously?

Reviewer #4: Yes

4. Have the authors made all data underlying the findings in their manuscript fully available (please refer to the Data Availability Statement at the start of the manuscript PDF file)?

Reviewer #4: Yes

5. Is the manuscript presented in an intelligible fashion and written in standard English?

Reviewer #4: Yes

Reviewer #4: The authors have clearly responded to feedback from previous reviews, demonstrating a strong commitment to scientific integrity and transparency. The revised manuscript presents the Dharma framework as both a powerful predictive tool and a practical clinical decision support system. By incorporating relevant medical knowledge into data handling and model design, the authors have linked complex machine learning with real-world healthcare needs.

1. The authors discussed how to referencing the study "COVID-19 Pandemic: A Worldwide Critical Review." This study demonstrates how mathematical models, expert advice, and computer simulations can be utilized to predict health issues and their outcomes. It also explains how to utilize a tool called Dharma to enhance predictions in healthcare.

2. Unlike many "black-box" AI models, Dharma is based on clinical reasoning. It employs a method for handling missing data that is grounded in clinical practice. This approach makes the model useful in various healthcare settings, especially those with limited resources.

3. Dharma demonstrates an impressive AUC-ROC of 0.98 and an accuracy rate of 93%, greatly outperforming traditional scoring systems like the Alvarado and Pediatric Appendicitis Scores.

4. The framework includes a model specifically for assessing the severity of conditions. The authors purposefully designed the model with a 5:1 class weight to ensure it has a high sensitivity (96%) for complicated appendicitis, prioritizing patient safety by reducing the chances of dangerous false negatives.

5. This research notably shows how the model works well even when the appendix is not visualized, a common limitation of ultrasound. Dharma maintains a strong AUC-ROC of 0.96 in these cases, providing a dependable option when imaging is unclear.

6. The use of SHAP (Shapley Additive Explanations) helps clinicians understand the reasons behind each prediction. This transparency is crucial for establishing trust in the use of AI for critical surgical decisions.

7. The authors have made further progress by launching Dharma as a free web application (dharma-ai.org) and sharing their framework on GitHub. This move promotes fairness in healthcare by providing robust diagnostic support to areas with limited resources.

**Do you want your identity to be public for this peer review?** For information about this choice, including consent withdrawal, please see our Privacy Policy

Reviewer #4: No
